# Gradient-mixing LEGO robots for purifying DNA origami nanostructures of multiple components by rate-zonal centrifugation

**Jason Sentosa**[1,2☯], **Franky Djutanta**[1,3☯]*, **Brian Horne**[4☯], **Dominic Showkeir**[4☯], **Robert Rezvani**[1,3], **Chloe Leff**[1,5], **Swechchha Pradhan**[1,3], **Rizal F. Hariadi**[1,5]*

**1** Biodesign Center for Molecular Design and Biomimetics (at the Biodesign Institute) at Arizona State University, Tempe, AZ, United States of America, **2** Department of Biomedical Engineering, Georgia Institute of Technology, Atlanta, GA, United States of America, **3** School for Engineering of Matter, Transport and Energy, Arizona State University, Tempe, AZ, United States of America, **4** Department of Physics, Arizona State University, Tempe, AZ, United States of America, **5** School of Molecular Sciences, Arizona State University, Tempe, AZ, United States of America

☯ These authors contributed equally to this work.
* fdjutant@asu.edu (FD); rhariadi@asu.edu (RFH)

**Data Availability Statement:** All the data supporting this study are provided within the paper and in https://zenodo.org/badge/latestdoi/446256259.

## Abstract

DNA origami purification is essential for many fields, including biophysics, molecular engineering, and therapeutics. The increasing interest in DNA origami has led to the development of rate-zonal centrifugation (RZC) as a scalable, high yield, and contamination-free method for purifying DNA origami nanostructures. RZC purification uses a linear density gradient of viscous media, such as glycerol or sucrose, to separate molecules according to their mass and shape. However, many methods for creating density gradients are time-consuming because they rely on slow passive diffusion. To expedite the preparation time, we used a LEGO gradient mixer to generate rotational motion and rapidly create a quasi-continuous density gradient with a minimal layering of the viscous media. Rotating two layers of differing concentrations at an angle decreases the time needed to form the density gradient from a few hours to minutes. In this study, the density gradients created by the LEGO gradient mixer were used to purify 3 DNA origami shapes that have different aspect ratios and numbers of components, with an aspect ratio ranging from 1:1 to 1:100 and the number of components up to 2. The gradient created by our LEGO gradient mixer is sufficient to purify folded DNA origami nanostructures from excess staple strands, regardless of their aspect ratios. Moreover, the gradient was able to separate DNA origami dimers from DNA origami monomers. In light of recent advances in large-scale DNA origami production, our method provides an alternative for purifying DNA origami nanostructures in large (gram) quantities in resource-limited settings.

## Introduction

DNA origami, a method of self-assembling complex nanostructures from long, single-stranded DNA (scaffold strands) and large excess of shorter oligonucleotides (staple strands) [1–8], has

**Funding:** The research in Hariadi lab was supported by the National Institutes of Health Director's New Innovator Award (1DP2AI144247), National Science Foundation SemiSynBio II (2027215), and Arizona Biomedical Research Consortium (ADHS17-00007401). The funders had no role in study design, data collection and analysis, decision to publish, or preparation of the manuscript.

**Competing interests:** The authors have declared that no competing interests exist.

proven to be a robust and efficient method for generating nanostructures with arbitrary shapes at ∼5 nm resolution. Furthermore, the exquisite positional control of DNA origami enables precise patterning of biomolecules and inorganic molecules [8–11]. The programmable DNA origami has been used in fields such as medicine [12–16], super-resolution microscopy [17–20] and electronics [21–23]. Although efforts have been made to optimize DNA origami assembly [24, 25], the yield of well-folded origami for complex 3D structures is far less than 100% [26]. Most applications of DNA origami, however, require uncontaminated samples that are free of staple strands and misfolded structures [26–28]. Thus, for downstream applications, a purification step is added after the folding step to isolate the well-folded structures.

Rate-Zonal Centrifugation (RZC) is a high-yield, contamination-free method for purifying DNA origami [27]. The technique subjects a linear density gradient to high centrifugal force in order to separate heterogeneous molecules by their distinct mass and shape [29]. Within the context of DNA origami, RZC can separate well-folded structures from other undesired species (misfolded structures, staple strands, and aggregates) as a purification step. RZC has several advantages compared to other purification methods, such as agarose gel electrophoresis (AGE). RZC keeps the samples in an aqueous solution throughout the process, is free from contaminants such as agarose gel residues, and is scalable to accommodate a large amount of sample [27]. In most cases, RZC purification requires fewer steps than multi-round spin filter purification or agarose gel extraction [30].

RZC purification starts with the traditional preparation of the density gradient, which can be time consuming [27, 31]. There are several methods to prepare a density gradient of glycerol: (1) layering two solutions of different glycerol concentrations in a tube and resting the tube horizontally for 1–2 hours to allow the glycerol to passively diffuse, (2) layering several solutions of different glycerol concentrations and resting the tube upright so that the layers can passively diffuse, and (3) mixing the glycerol gradient using a commercially available gradient mixer. Since the first two methods employ passive diffusion, they do not require expensive equipment but come with a lengthy preparation period. Utilizing a gradient mixer reduces the preparation time, but it is costly and may require additional equipment or training. Thus, there is a need for a method for preparing a density gradient that is both fast and cost-effective. Here, we report a low-cost method for creating a linear glycerol density gradient that accelerates diffusion via a simple rotational motion. The prototype of the instrument was made using LEGO EV3 Mindstorms materials, and the diffusion process facilitated by the LEGO gradient mixer takes only one minute.

To develop the gradient, a low-concentration solution of glycerol is gently layered on top of a high-concentration glycerol solution inside a centrifuge tube. The glycerol-filled tubes are then loaded into the LEGO gradient mixer and the program is initiated to tilt the glycerol-filled tubes horizontally and rotate them for 1 minute at low rpm (20 rpm). The LEGO gradient mixer then returns to its upright position, yielding the density gradients for the RZC purification. The DNA origami samples are loaded on top of the newly created glycerol gradients. The tubes are then transferred to an ultracentrifuge for RZC purification [27]. Fractions are collected from the RZC result to be analyzed using AGE, and the purified structures are verified using atomic force microscopy (AFM).

## Materials and methods

### Buffers, reagents, materials and equipments

All buffers, reagents, materials, and equipment can be found in Section S3 of the S1 File.

## Design of the gradient mixer

All components for building the LEGO gradient mixer are part of the EV3 LEGO Mindstorms kit (Item no. 31313) except the ultracentrifuge tube holder (Fig 1(1)), which was 3D printed using an Ultimaker 3 3D printer (part no. 9671) (see S1 Fig in S1 File for building instruction). The design features two motors: a spinning motor (Fig 1(2)) and a turning servo motor (Fig 1 (3)). The centrifuge tube holder is connected to the spinning motor, which is connected to the servo motor by the large gray gear (Fig 1(4)). Both motors are supported by the scaffold (Fig 1 (5)), which connects directly to the large gray gear and allows the spinning motor and the centrifuge tube holder to rotate into a horizontal position. Both motors are connected to a LEGO Brick (Fig 1(6)) programmed with the mixing protocol. When the LEGO gradient mixer is ready to be used, the servo motor slowly tilts the tubes into a horizontal position (from Fig 1A and 1B), increasing the surface area between the glycerol solutions and accelerating diffusion. After reaching its horizontal configuration, the spinning motor begins to rotate the tubes. This allows layers of different concentrations to diffuse quickly with a smooth transition. Once the spinning motor stops, the servo motor returns the tubes back to their initial position, and the gradient is ready for RZC.

## DNA origami design and assembly

The design for 6-hb monomers and dimers was adopted from Shawn Douglas's paper [32]. The monomer was constructed as a single DNA origami while the dimer was constructed from two separate pieces of precursor monomer (left and right). The left and right precursor monomers were annealed individually with different staple strands such that one side of each precursor monomer was on and the other side was off. The resulting left and right monomers were filtered twice using a 100 kD amicon filter spun at 4,500 ×g for 5 mins in order to remove excess staples that may have inhibited dimerization. The filtered products were then combined into a single tube to dimerize overnight. The CADnano file for both 6-hb monomers and dimers can be found in the S1 File.

The DNA nanostructures were assembled in a one-pot reaction by mixing 30 nm p8064 scaffold strands and 300–900 nm staple oligonucleotides (S1 Table in S1 File) in a buffer solution containing $1 \times$ TAE 12.5 mM $MgCl_2$. The mixture was then annealed using a thermocycler that gradually cooled the mixture from 90˚C to 30˚C over 1.5 hours. The predicted length of a 6-hb monomer was determined to be 460 nm for 32 segments of 42 bp each, assuming that the length of each base pair is 0.34 nm. The dimers were predicted to have a length of 920 nm with 32 segments from the left and right parts and ⅔ segment for connectivity (total 64⅔ segments) of 42 bp for each segment. The diameter of the 6-hb origami was calculated assuming an interhelical distance of 3 nm [33] is approximately 8 nm.

## Preparation of the glycerol gradient

A 15–45% (concentrations depend on the origami of interest) quasi-continuous glycerol gradient was prepared by pipetting 300 µL of 45% glycerol (v/v) in 1× TAE 12.5 mM $MgCl_2$ buffer into a 1.0 mL centrifuge tube. The same volume of 15% glycerol solution was then carefully layered on top of the 45% glycerol using either a dropper or a syringe to prevent disturbing the surface of 45% glycerol (Fig 2A.1). A clear division between the two solutions was visible after the layering steps. The centrifuge tubes were then inserted into the LEGO gradient mixer and mixed for 1 minute, which was sufficient for the 1.0 mL tubes (Fig 2A.2–2A.4). A quasi-continuous density gradient was visible after mixing with the LEGO gradient mixer.

The ideal spin time was determined using a visual test. First, the 45% glycerol solution was dyed with blue food coloring. Both the clear 15% glycerol and the colored 45% glycerol

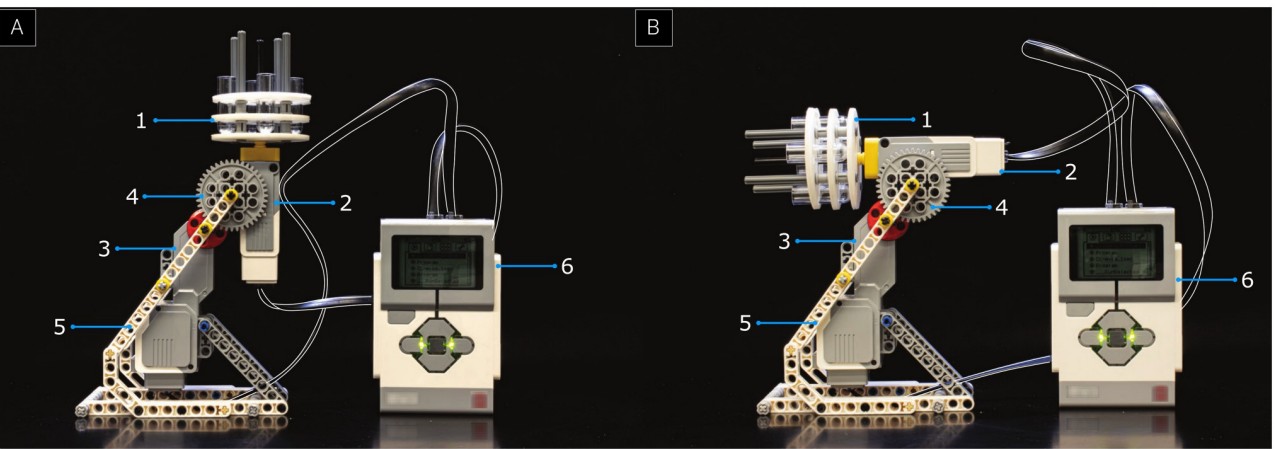

**Fig 1.** Side view of the LEGO gradient mixer during (**A**) its initial position and (**B**) its horizontal tilting phase. (**1**) 3D printed centrifuge-tube holder. (**2**) Spinning motor to rotate the tubes while in horizontal position. (**3**) Turning servo motor responsible for tilting the tubes horizontally. (**4**) Large grey gear connecting the two motors with its small gear complement. (**5**) The scaffold holding the structure together. (**6**) The LEGO controller for orchestrating the motions of the two motors. The black cables are traced in white for clarity.

(300 μL each) were added to the 1.0 mL centrifuge tubes (Fig 2B). The tubes were loaded into the mixer and different spin times were tested until a smooth transition from clear (15%) to blue (45%) was visible (Fig 2C). Three other glycerol gradient concentrations (30%/45%, 15%/60%, 30%/60%) were also tested with spin times of 30, 60, 120, and 180 sec. (Fig 3). The detailed protocol programmed in the LEGO gradient mixer for both the experiment and the visual test can be found in the S1 File.

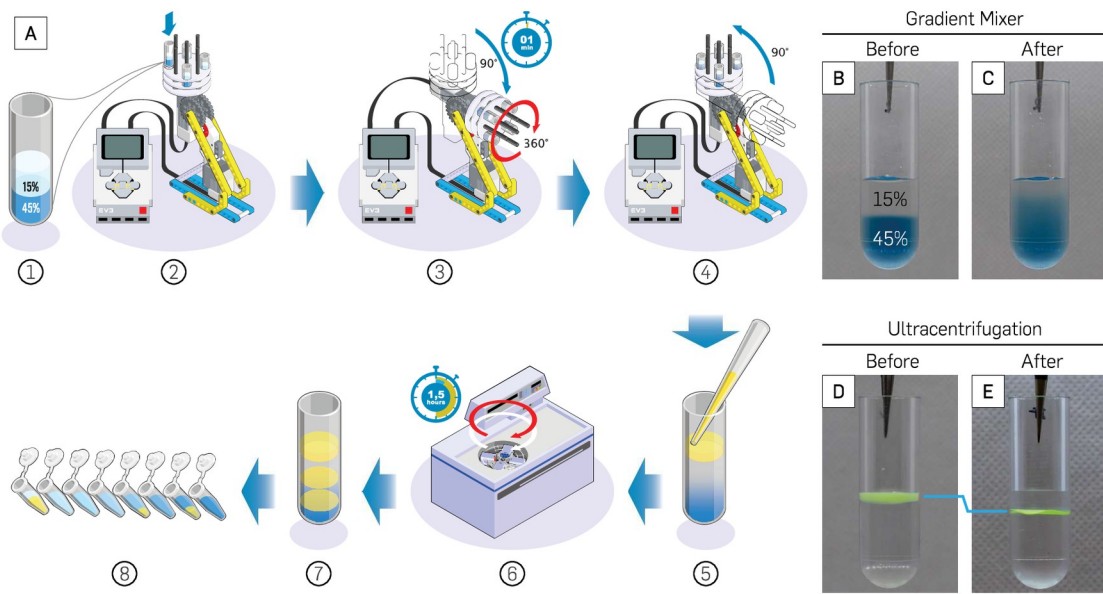

**Fig 2. Glycerol gradient preparation and RZC purification (S1 Movie in S1 File).** (**A**) Preparation of glycerol gradient with LEGO gradient mixer and separation of sample by RZC. (**B**) Layers of glycerol before mixing; the blue glycerol at the bottom is 45% (v/v) and the clear glycerol at the top is 15% (v/v). (**C**) Linear glycerol gradient after mixing with LEGO gradient mixer. (**D**) 140 nm green fluorescent beads on top of the glycerol gradient before RZC. (**E**) Fluorescent beads after RZC concentrated into a thin layer due to separation from the glycerol gradient. All glycerol solutions were diluted in 1× TAE 12.5 mM MgCl$_2$.

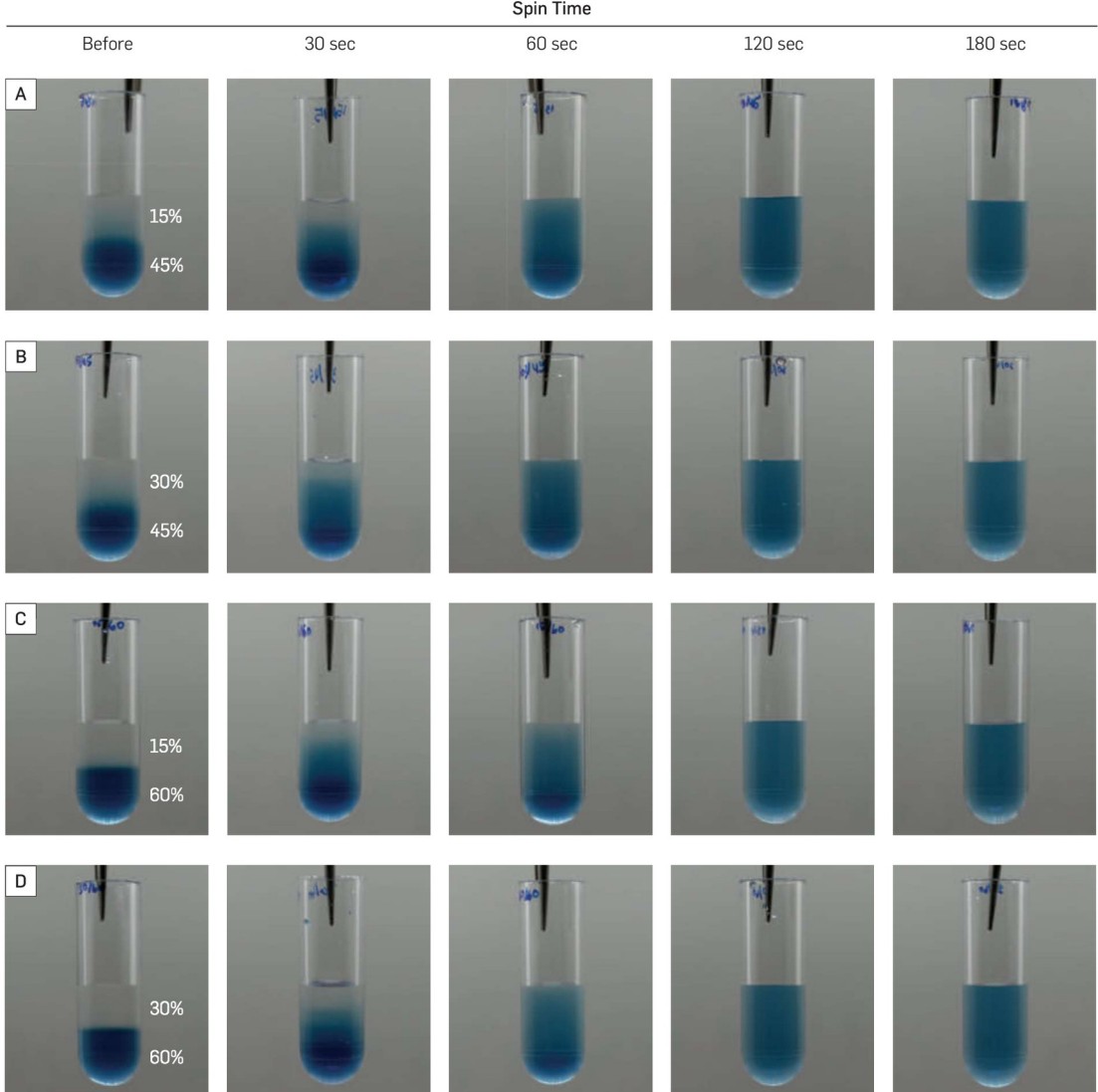

**Fig 3. Comparison of different spin time for different gradients.** (A–D) Four different combinations of glycerol concentrations before and after loading into the LEGO gradient mixer. The bottom layer of glycerol is dyed blue for visual inspection, while the top layer is left clear. Four spin times were selected to compare the glycerol gradient formed at the indicated spin times. As can be seen by the color gradient created as the blue-dyed glycerol mixes with the clear glycerol, each gradient, and spin time combination lead to comparable results.

### Centrifugation and sample extraction procedure

After the glycerol gradient was formed, 50–100 μL of the DNA origami sample containing 10% glycerol was loaded on top of the gradient by hovering the pipette tip near the surface of the gradient (Fig 2A.5). Most of the sample should rest on the surface of the gradient. The centrifuge tubes containing the sample and gradient were loaded into a swinging bucket rotor (Beckman TLS 55). The samples were centrifuged at 50,000 rpm ($\sim$ 150,000 g) for 1.5 hours at 4°C (Fig 2A.6). For different origami, the exact RPM and the duration of centrifugation need to be optimized. Once the centrifugation was complete, 16 to 18 equal-volume fractions of the sample (Fig 2A.7 and 2A.8) were then collected from bottom to top using longneck gel-loading

tips to preserve the glycerol layers (see S1 Movie in S1 File for full instructional video from gradient preparation to fractionation). Fluorescent beads (G140) were used as a control sample for RZC protocol. The beads were laid on top of the 15–45% glycerol gradient (Fig 2D) to be centrifuged with the above protocol. The resulting tube showed the beads concentrating in a thin layer that lies below its initial position on the gradient's surface (Fig 2E).

## Experiment calibration

The gradient parameters and centrifugation setup were optimized using a SYBR Gold-stained sample that allowed for a quick assessment of separation. The sample being purified was stained with 3× SYBR Gold prior to loading onto the surface of the gradients. The centrifuge tubes containing the stained sample and gradient were centrifuged at 50,000 rpm ($\sim$150,000 g) for 1.5 hours at 4°C. After centrifugation, the tubes were visualized with Invitrogen Safe Imager 2.0 Blue Light transilluminators, and the images were taken with a Canon lens EF-S 35mm f/2.8 Macro IS STM covered with a single B+W 49mm XS-Pro clear MRC-nano 007 filter on a Canon EOS 77D camera body at 30-sec exposure time triggered by a smartphone to avoid vibration. The target, properly folded structures were visible as a thin horizontal band on the gradient while the staple strands were visible as a cloudy region near the top of the gradient. The control experiment for this method using SYBR gold-only sample was performed in a 30–60% (v/v) glycerol gradient (see S2 Fig in S1 File for result). The optimum parameters were determined by conditions which provided the greatest separation between the structures being purified and the undesired molecules.

## Result confirmation and analysis

After the centrifuged sample gradient was partitioned into fractions, 10 μL from each fraction was taken and mixed with 1 μL of 20× SYBR gold and 1 μL of loading dye. The fractionated samples were analyzed using 1% Agarose Gel at 75 V for 1.5 hours. Once the electrophoresis was complete, the gel was post-stained with 1× SYBR gold and imaged using ultraviolet light in Bio-Rad Molecular Imager Gel Doc XR+ (1708195EDU). The fractions containing the desired DNA origami were then combined into a single tube for buffer exchange. The purpose of the buffer exchange was to ensure that the DNA nanostructures returned to their native buffer without any glycerol.

The result of gradient purification was also confirmed by comparing the purified and unpurified samples with AFM imaging. Both purified and unpurified samples were diluted by 100× with 1× TAE 12.5 mM $MgCl_2$ to ensure the DNA nanotubes were sparsely distributed in the AFM images. After preparing and calibrating the AFM, four samples were imaged and compared: unpurified 6-hb monomer, purified 6-hb monomer, unpurified 6-hb dimer, and purified 6-hb dimer.

## Statistical analysis

The staple strands and DNA origami were distinguishable by their shape and intensity in the AFM images. While the 6-hb monomer and dimer were distinguishable by their length in the AFM images, $\sim$500 nm for a monomer and $\sim$1000 nm for a dimer. The AFM images were processed in imageJ and Gwyddion. The percentage of staple strands percentage was calculated with a cumulative distribution function (CDF) fit on the pixel intensity of the AFM images. CDFs were fitted to three groups: AFM surface, staple strands, and 6-hb monomers. The resulting optimized parameters for each group determine the percentage of pixels associated with each group. The CDF analysis was performed in Mathematica and the analysis script can be found in the S1 File. Meanwhile, the number of monomers and dimers was manually

counted to calculate the dimer content of both the unpurified and purified dimers. The bootstrap method was used to calculate the standard error of the mean of the monomer and dimer content. The bootstrap calculation was performed with Mathematica. First, the monomer and dimer were assigned as binary numbers 0 and 1, respectively. Then, from the full sample data set *N*, a subset of size *N/2* was randomly chosen to compute the mean. The size of the subset was rounded to the nearest integer, and an element was never chosen more than once. 10,000 subsets and means were generated with randomly chosen elements each time. Finally, the standard deviation of the means was used to estimate the standard error of the monomer and dimer content.

## Results

In this study, three DNA origami structures, namely 6-helix budle (6-hb) DNA nanotube monomers (Fig 4A), 3D DNA origami snub cubes (Fig 4F), and 6-hb DNA nanotube dimers (Fig 5A), were purified using the glycerol gradient created by the LEGO gradient mixer. The effectiveness of the gradient used for rate-zonal centrifugation (RZC) was evaluated using gel electrophoresis and AFM imaging. The sample separation after fractionation between staple strands, misfolded structures, and well-folded structures was compared to the results of RZC using other glycerol gradient preparation methods [27].

### Ideal glycerol gradient and spin time

We first systematically tested different glycerol gradients with varying spin times (Fig 3). Within our experimental parameters, we found that the 60-sec spin time was sufficient to create a smooth glycerol gradient. A 30-sec spin time produced a gradient that is closer to a step gradient than a smooth gradient, while longer spin times (120- and 180- sec) yielded a near-uniform solution rather than a gradient. Surprisingly, all four glycerol concentration combinations (Fig 3A–3D) produced smooth glycerol gradients after the spin time of 60 seconds. The consistency of the gradient produced was tested by calculating the deviation from the plot profiles of 20 gradients created by the LEGO gradient mixer using the same parameters (S3 Fig in S1 File). The absolute deviation from the mean was calculated to be 4.9±4.7% (mean±SD, *N*=20), indicating that the gradient production of the LEGO gradient mixer is highly reproducible.

### Excess staple strands remain in the top section of the sample after ultracentrifugation

The AGE results (Fig 4B) showed that the well-folded structures were contained in 10–20% of the volume of the glycerol gradient and most of the staple strands accumulated near the top of the glycerol gradient, considerably separated from the origami of interest. AFM images of the unpurified and purified 6-hb monomers revealed that most of the staple strands were removed after purification (Fig 4C and 4D and S4 Fig in S1 File). The mica surface of the AFM image of the unpurified monomer was crowded with excess staple strands that reside in the mixture, as represented by the grainy background surface (Fig 4C). On the other hand, the AFM image of the purified monomer consistently gives less background surface (Fig 4D). The SYBR gold-stained 6 hb monomer sample also showed separation between well-folded monomer and excess staple bonds after RZC purification using a glycerol gradient concentration of 15% to 45% with 50k rpm ultracentrifugation at 4˚C for 1.5 hours (Fig 4G).

Image analysis of the staples strands in the AFM images demonstrated a decrease in staple strands content. To quantify the remnant of the staple strands in fractions 14 and 15 in Fig 4B, we modeled the AFM pixel height distribution data as the sum of 3 Gaussian functions (black)

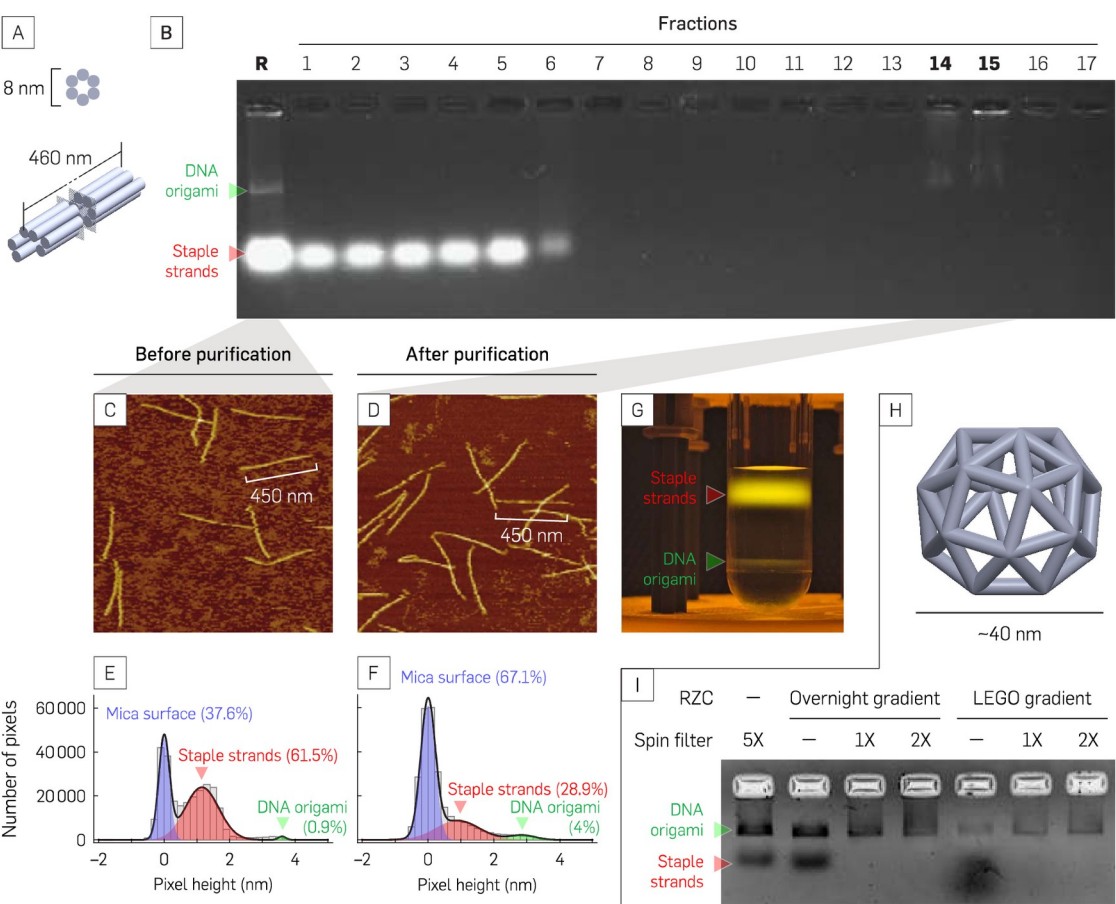

**Fig 4. RZC purification of high and low aspect ratio DNA origami nanostructures.** (**A**) 3D rendered model of high aspect ratio 6-helix bundle DNA origami nanotube (6-hb) monomers. (**B**) Gel result of liquid fractions from top to bottom (fractions 1–17) of the 6-hb sample. *R* is unpurified samples as a negative control lane. Fractions 14 and 15 correspond to the fractions containing the purified monomers. (**C**–**F**) AFM images and their corresponding heigh distribution of 6-hb monomer before (**C** and **E**) and after (**D** and **F**) RZC purification. (**G**) RZC purified 6-hb monomer SYBR gold stained 6-hb monomer sample after RZC purification. (**H**) 3D rendered model of a 40-nm DNA origami snub cube (SC). (**I**) Gel shift assays for indicated RZC-purified SC samples using glycerol gradients prepared by overnight passive diffusion or LEGO gradient mixer, followed by ultrafiltration.

corresponding to 3 groups, which are the mica surface (blue), staple strands (red), and DNA origami nanostructures (green).

$$n(z) \propto P_{mica} \frac{1}{\sigma_{mica}} e^{-\frac{1}{2}\left(\frac{z - \mu_{mica}}{\sigma_{mica}}\right)^2} + P_{staple} \frac{1}{\sigma_{staple}} e^{-\frac{1}{2}\left(\frac{z - \mu_{staple}}{\sigma_{staple}}\right)^2} + P_{origami} \frac{1}{\sigma_{origami}} e^{-\frac{1}{2}\left(\frac{z - \mu_{origami}}{\sigma_{origami}}\right)^2} \tag{1}$$

The cumulative distribution of pixel heights was fitted to the cumulative distribution function (CDF) of Eq 1 using the following 9 fitting parameters.

$P_{mica}$ = fraction of pixels that corresponds to mica surfaces

$P_{staple}$ = fraction of pixels that corresponds to staple strands

$P_{mica}$ = fraction of pixels that corresponds DNA origami nanostructures

$\mu_{mica}$ = mean height of mica surfaces

$\mu_{staple}$ = mean height of staple strands

$\mu_{mica}$ = mean height of DNA origami nanostructures

$\sigma_{mica}$ = standard deviation of the observed height of mica surfaces

$\sigma_{staple}$ = standard deviation of the observed height of staple strands
$\sigma_{mica}$ = standard deviation of the observed height of DNA origami nanostructures
and with a normalization constraint below

$$P_{mica} + P_{staple} + P_{origami} = 1$$

Based on the fit, the inferred percentages of pixels corresponding to the staple strands ($P_{staple}$) is 62% in monomer unpurified (Fig 4E; $N$ = 117 DNA origami) and 29% in monomer purified (Fig 4F; $N$ = 281 DNA origami). After RZC purification, $P_{staple}$ was reduced by 2.14×, further showing that RZC removed most of the excess staple strand.

## 6-hb monomer structure was preserved after purification

Intact purified DNA origami is critical for downstream applications. Inspection of DNA origami using AFM revealed that the monomer structures were not compromised (Fig 4D) by the purification. The 6-hb monomer was neither degraded nor physically altered after the purification step. The AFM images also showed the absence of aggregation before and after RZC purification. For some downstream applications, buffer exchange may be required for long-term storage of the structures. Alternatively, sucrose can be used to substitute for glycerol. A sucrose gradient created using the LEGO gradient mixer was able to separate the dimer with effectiveness similar to the glycerol gradient (see S5 Fig in S1 File for dimer separation using the sucrose gradient).

## LEGO gradient mixer purified low aspect ratio snub cube DNA origami

DNA origami of complex structure, such as a snub cube (Fig 4H), was also purified using the LEGO gradient mixer RZC method. The AGE result in Fig 4I shows a comparison of the snub cube purification methods. In lane 1, the snub cube was purified via 5 rounds of centrifugation in a 100 kDa Amicon spin column to remove excess staple strands from the annealed DNA origami structure. In the rest of the lanes, the snub cubes were purified with RZC. Lanes 2 through 4 were purified with RZC using a gradient prepared by layering 7 different concentrations of glycerol (from 15% to 45% top to bottom with a 5% difference between each new layer) and incubating the tube overnight at room temperature. Meanwhile, the glycerol gradients used in lanes 5 through 7 were made using the LEGO gradient mixer and two initial layers with 15% and 45% glycerol concentration. In both methods, after using the glycerol gradient to perform RZC on the snub cube, a buffer exchange was carried out using a 100 kDa Amicon spin column to remove glycerol from the native 1× TAE 12.5 mM MgCl$_2$ buffer. After RZC and an initial AGE result, the fractions determined to contain purified origami were pooled into two tubes for each gradient preparation method that then underwent buffer exchange. The results of these buffer exchanges are represented by lanes 3 and 4 for the overnight diffusion method and lanes 6 and 7 for the LEGO gradient mixer method.

Comparisons between methods for forming glycerol gradients showed that the LEGO mixer gradient resulted in a higher yield than the overnight diffusion gradient after RZC. The yield was determined by an absorbance measure of the combined fractions containing the purified snub cube, as indicated by the AGE result (Fig 4I, not the fractionation AGE result in Fig 4B). Taking the combined volume of the purified samples of multiple fractions, the yield for DNA purification using 5× spin-filter, overnight gradient, and LEGO gradient were measured to be 1.98, 1.28, and 1.41 pmol, respectively ($N$=1). Although the yield of the purified samples using the gradient produced by the LEGO mixer (lanes 5–7) was higher than that of the purified samples using the overnight diffusion gradient (lanes 2–4), neither the RZC method gave a significantly higher yield than purification using traditional spin filters (lane 1).

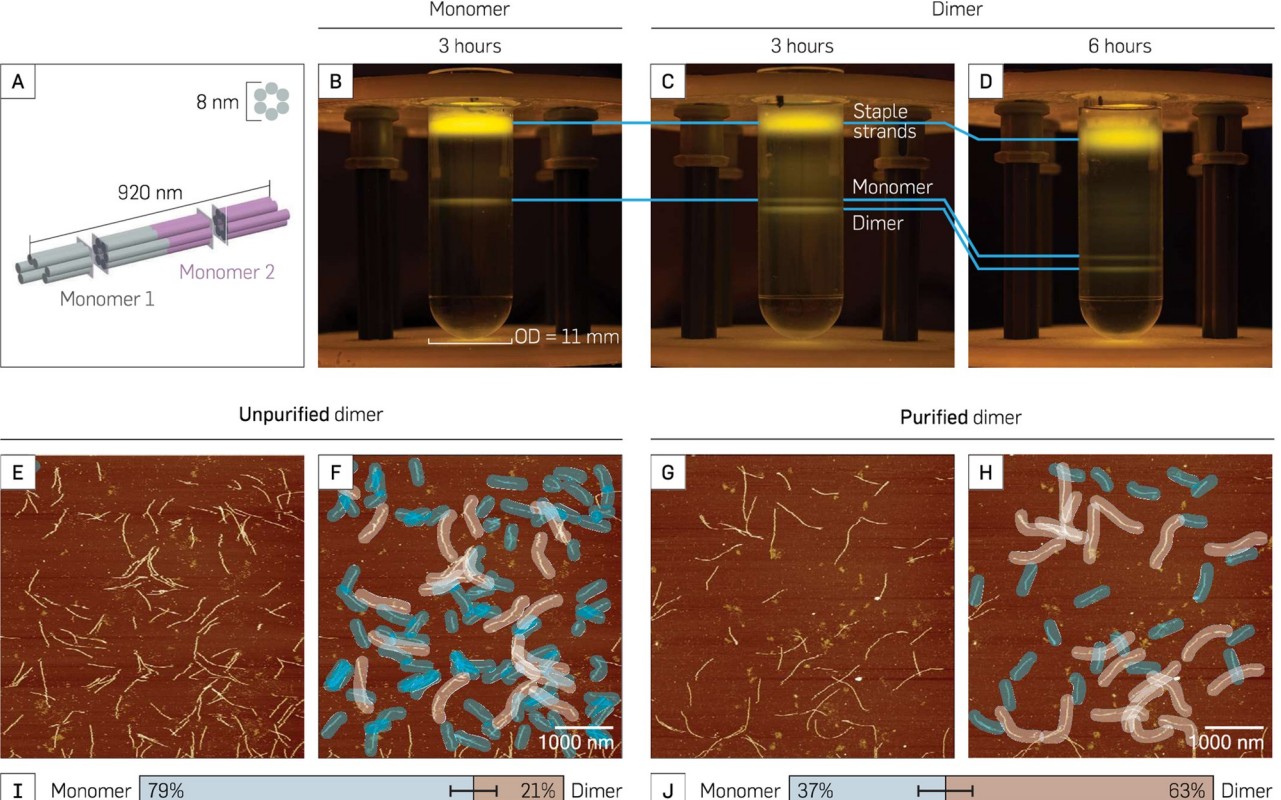

**Fig 5. RZC purification of 6-hb dimer.** (**A**) 3D-rendered model of a 6-hb dimer. (**B**) Comparison between SYBR gold stained 6-hb monomer (left) and (**C**) 6-hb dimer (right) purified using a 30%—60% gradient column centrifuged for 3 hours at 50k rpm in 4°C. (**D**) SYBR gold stained dimer with 6 hours of centrifugation. (**E**) AFM image of unpurified dimer. (**F**) Precursor monomer (highlighted blue) and dimer (highlighted light brown) from unpurified dimer. (**G**) AFM image of RZC purified dimer. (**H**) Significantly smaller number of monomer (highlighted blue) with similar number of dimer (highlighted light brown) compared to those in unpurified dimer. (**I** and **J**) Comparison of precursor monomers (highlighted blue) and dimer (highlighted light brown) between unpurified and purified dimer. Dimer content of unpurified dimer and purified dimer at 21±5% and 63±6% respectively. Standard deviation was calculated with the bootstrapping method (*N*=273 for unpurified dimer; *N*=114 for purified dimer).

## 6-hb dimers were purified from its monomers

The separation of the dimer from its precursor monomer using the gradient formed by the LEGO gradient mixer was largely successful (Fig 5, more AFM images in the S1 File). Initial attempts using the same parameters as in Fig 4 with the 6-hb dimer did not show separation between the dimer and its precursor monomer. Optimization of the experimental conditions for the 6-hb dimer was accomplished with the help of an SYBR gold staining. It was determined that a glycerol gradient concentration of 30% to 60% (600 μL each) spun at 50,000 rpm at 4°C was able to separate the dimer and its precursor monomer. The dimer was compared to the monomer in the same gradient condition after 3 hours of centrifugation at 50,000 rpm at 4°C (Fig 5B and 5C).

The monomer separation resulted in a single thin DNA origami band that corresponded to the 6-hb monomers. On the other hand, the purification of 6-hb dimers yielded two bands: the upper band of the precursor monomers and the lower band of the 6-hb dimers. The upper band migrated approximately the same distance down the gradient as the band present in the 6-hb monomer sample, further showing that the upper band corresponds to the precursor monomers. Because the dimer has twice the mass and size of the monomer, the dimer

migrated further than the monomer during RZC. As expected, longer ultracentifugation time resulted in a greater separation of DNA origami monomer and dimer bands. Comparison between the unpurified and purified dimer AFM images showed that there are > 2× more monomeric structures in the unpurified dimer (79±5%; Fig 5E and 5F) than in the purified dimer (37±6%; Fig 5G and 5H). RZC was able to filter out the monomer, while still retaining a similar number of dimers before and after the purification step. The observed monomer in the purified dimer fraction (Fig 5G–5J) was likely due to fragmentation induced by fluid flow and accidental mixing during the extraction of the monomer and dimer fractions with a micropipette tip with a long neck pipette tip.

## Discussion

DNA nanotechnology has the potential to advance research in the fields of microscopy [17–20], nanomedicine [12–16], and even molecular-scale electronics [21–23]. Its pace toward industry applications is paired with successful efforts to remove the key obstacle of scalability for both production [34–36] and purification [27] of DNA nanostructures. The LEGO gradient mixer presented here is an effective complement to the current state-of-the-art RZC method to purify DNA origami. The LEGO gradient mixer was able to produce a concentration gradient in a relatively short time compared to other gradient mixers [27, 31]. In combination with the speed and ease of use of the LEGO gradient mixer, RZC purification could help pave the way for large-scale DNA nanostructure production that will potentially enable the separation of functional DNA nanostructures from the side products of bioconjugation reactions, such as excess streptavidins, quantum dots, gold nanoparticles, aptamers, and other moeties. The gradient produced by this method was able to purify 6-hb monomer from its staple strands and 6-hb dimer from its precursor monomer, demonstrating its ability to separate molecules with 1:2 size ratio. Although the purified dimer shows only 63±6% purity (Fig 5J), the observed monomer in the purified dimer likely results from the difficulty of extracting the dimer sample manually from the small gradient volume in which the experiment is conducted. Taking note that we used less than ideal equipment for the purification method, *i.e.* long neck gel loading tips, the yield could be significantly improved by upgrading or automating the fractionation technique. An improved and/or automated fractionation technique could produce more consistent and higher yield results and would therefore be integral to optimizing the effectiveness of RZC purification. Similar success was observed when the RZC purification and LEGO mixed gradient were applied to the purification of a DNA origami snubcube (Fig 4H). Though the Amicon spin column purification performed better in purifying the snubcube from its staple strands, the purification method using the LEGO mixed gradient offered comparable results. Translating the success of RZC from a more simplified 6-hb monomer and dimer to a more complex shape, like the snubcube, makes RZC a promising purification method for a wide range of DNA origami products. The strength of RZC is its ability to purify molecules that require finer precision, such as separating a 6-hb dimer from its monomer.

We note that the reported method can be further improved to reduce the cost. The current cost of the EV3 LEGO Mindstorms pieces used to make this device is $349.99, while gradient mixers are typically quoted at around $500.00 to $650.00 for a Millipore Sigma gradient mixer (cat no. Z340391) and >$2K for a GradientMaster (Biocomp Instruments). The main components of the LEGO gradient mixer are a motor, a servo, and a custom holder for the tubes, all of which do not require a high degree of precision to be effective. Hence, the price to create such a LEGO gradient mixer can be reduced further by an order of magnitude (S2 Table in S1 File). To lower the cost further, such machines can be powered manually using some gear and scaffolding materials. In the spirit of frugal science, our discovery could help density gradient

centrifugation become a more accessible tool for scientific communities around the world. The myriad applications of density gradients to fractionate DNA origami, viral particles, and a variety of macromolecules [37] have been critical in the preparation of viruses for vaccines and other immunotherapy products. The methods introduced in this paper will contribute to science education and research in resource-limited settings.

## Supporting information

**S1 File. This document provides comprehensive supporting information, including detailed protocols, materials, equipment, 5 supplementary figures, 2 supplementary tables, and 1 supplementary movie.**
(PDF)

## Acknowledgments

The work was initiated by a final project in PHY 472 (Advanced Biophysics Laboratory) in the Department of Physics at Arizona State University. We acknowledge students in PHY 472, Carter Swanson, Adrian Kwiatkowski, and Aliyapadi Biruni Hariadi for valuable discussions. The authors thank Sapto Cahyono for assistance with illustrations, Evangeline Taylor-Hermes for pointing out the LEGO software for the building instructions, and Gde Bimananda Wisna Mahardika for technical assistance. AFM images were collected in the laboratory of Hao Yan at Arizona State University. The authors also thank Jeffrey La Belle's group for lending us the rotor used for ultracentrifugation.

## Author Contributions

**Conceptualization:** Brian Horne, Dominic Showkeir, Rizal F. Hariadi.

**Data curation:** Jason Sentosa, Franky Djutanta, Brian Horne, Dominic Showkeir, Robert Rezvani, Chloe Leff, Swechchha Pradhan.

**Formal analysis:** Jason Sentosa, Brian Horne, Dominic Showkeir, Robert Rezvani, Chloe Leff, Swechchha Pradhan, Rizal F. Hariadi.

**Funding acquisition:** Rizal F. Hariadi.

**Investigation:** Jason Sentosa, Franky Djutanta, Robert Rezvani.

**Methodology:** Franky Djutanta, Rizal F. Hariadi.

**Project administration:** Rizal F. Hariadi.

**Resources:** Rizal F. Hariadi.

**Software:** Jason Sentosa, Rizal F. Hariadi.

**Supervision:** Franky Djutanta, Rizal F. Hariadi.

**Visualization:** Jason Sentosa, Franky Djutanta, Chloe Leff, Swechchha Pradhan, Rizal F. Hariadi.

**Writing – original draft:** Jason Sentosa, Franky Djutanta.

**Writing – review & editing:** Jason Sentosa, Rizal F. Hariadi.

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
