## [Decision Letter · Decision Letter 0]

27 Apr 2022

PONE-D-22-01705Gradient-mixing LEGO robots for purifying DNA origami nanostructures of multiple components by rate-zonal centrifugationPLOS ONE

Dear Dr. Hariadi,

Thank you for submitting your manuscript to PLOS ONE. After careful consideration, we feel that it has merit but does not fully meet PLOS ONE’s publication criteria as it currently stands. Therefore, we invite you to submit a revised version of the manuscript that addresses the points raised during the review process.

Please revise this manuscript.

We look forward to receiving your revised manuscript.

Kind regards,

Yuliang Zhang, Ph.D.

Academic Editor

PLOS ONE

Journal Requirements:

 “The research in Hariadi lab was supported by the National Institutes of Health Director’s New Innovator Award (1DP2AI144247), National Science Foundation SemiSynBio II

(2027215), and Arizona Biomedical Research Consortium (ADHS17-00007401).”

“The research in Hariadi lab was supported by the National Institutes of Health Director’s New Innovator Award (1DP2AI144247), National Science Foundation SemiSynBio II (2027215), and Arizona Biomedical Research Consortium (ADHS17-00007401).”

“The research in Hariadi lab was supported by the National Institutes of Health Director’s New Innovator Award (1DP2AI144247), National Science Foundation SemiSynBio II

(2027215), and Arizona Biomedical Research Consortium (ADHS17-00007401).”

5. Please amend the manuscript submission data (via Edit Submission) to include author Robert Rezvani.

Additional Editor Comments:

Few suggestions:

In Fig. 1, please adjust the background to show black wires or modify the color of the wires; change a, b, … ,f to numbers, similar to Fig. 2,Instead of building instruction, generate an animation for the assembly of the Gradient Mixer.The authors mentioned that Fig. 4 C and D demonstrated a significant difference between unpurified and purified samples. However, the quantitative analysis, such as surface roughness of background area in AFM images, will be more convincing than visual inspection.Regarding blot/gel data: PLOS ONE now requires that submissions reporting blots or gels include original, uncropped blot/gel image data as a supplement or in a public repository. This is in addition to complying with our image preparation guidelines described at https://journals.plos.org/plosone/s/figures#loc-blot-and-gel-reporting-requirements. These requirements apply both to the main figures and to cropped blot/gel images included in Supporting Information. On page 5, there is a typo “applicatios”.Please provide the sample size (total numbers of images or molecules) in the statistical analysis.

Reviewers' comments:

Reviewer's Responses to Questions

**Comments to the Author**

1. Is the manuscript technically sound, and do the data support the conclusions?

Reviewer #1: Partly

Reviewer #2: Yes

2. Has the statistical analysis been performed appropriately and rigorously? 

Reviewer #1: N/A

Reviewer #2: Yes

3. Have the authors made all data underlying the findings in their manuscript fully available?

Reviewer #1: Yes

Reviewer #2: Yes

4. Is the manuscript presented in an intelligible fashion and written in standard English?

Reviewer #1: Yes

Reviewer #2: Yes

5. Review Comments to the Author

Reviewer #1: The Jason et al. manuscript describes good equipment which can be well compact with the rate-zonal centrifugation in purifying DNA. The LEGO robots they made are very easily assembled and generate the critical point of purification (glycerol gradient) very high efficiency. Overall, this is a good method paper to make the new equipment. However, purifying DNA (RZC) is not new and well developed. Therefore, the insight point for this paper is to complete a smooth gradient of glycerol for RZC.

I still have some questions about this robot:

1. The authors made this LEGO robot with many Lego pieces and wrote the control code. And the function of this LEGO robot is to flip 90 degrees and rotate for a particular time. There are a lot of simple motors which can spin the samples. What is the advantage of this machine?

2. In figure 5D, the difference between monomer and dimer is apparent, and there is a big gap between them. Why are there 37% of the monomer in the purified dimer products? I ask this because sometimes the purity of the target product is critical and 63% of the purity is not high. If the difference between monomer and dimer is similar in the agarose gel, you can make the purity 100%.

Some minor comments:

1. In figure 2, the authors used glycerol gradient but used sucrose in the movie.

2. Does vision only distinguish the gradient of different glycerol in Fig.3? Are there any measurable parameters to quantify the gradient? Are these results repeatable very well each time?

Reviewer #2: In this work, the authors made LEGO robots to prepare RZC for purifying DNA origami nano-structures. They tested three kinds of DNA origami structures with different shapes and strand ratios. The results look promising and show the effectiveness of the robot’s gradient-mixing RZC. Although it is not a large-load work, it is interesting and inspires the perspective of nanostructure synthesis with robots. I am just curious how well the robot can be programmed.

6. PLOS authors have the option to publish the peer review history of their article (what does this mean?). If published, this will include your full peer review and any attached files.

Reviewer #1: No

Reviewer #2: No

---

## [Author Response · Author response to Decision Letter 0]

13 Feb 2023

Our detailed response to the reviewer's and editor's comments is in the uploaded Cover Letter.

---

## [Editor Report · Decision Letter 1]

3 Mar 2023

Gradient-mixing LEGO robots for purifying DNA origami nanostructures of multiple components by rate-zonal centrifugation

PONE-D-22-01705R1

Dear Dr. Hariadi,

We’re pleased to inform you that your manuscript has been judged scientifically suitable for publication and will be formally accepted for publication once it meets all outstanding technical requirements.

Kind regards,

Yuliang Zhang, Ph.D.

Academic Editor

PLOS ONE
---

## [Editor Report · Acceptance letter]

8 Jun 2023

PONE-D-22-01705R1 

Gradient-mixing LEGO robots for purifying DNA origami nanostructures of multiple components by rate-zonal centrifugation 

Dear Dr. Hariadi:

I'm pleased to inform you that your manuscript has been deemed suitable for publication in PLOS ONE. Congratulations! Your manuscript is now with our production department. 

Kind regards, 

on behalf of

Dr. Yuliang Zhang 

Academic Editor

PLOS ONE